# Evaluation of Heterocyclic Carboxamides as Potential Efflux Pump Inhibitors in *Pseudomonas aeruginosa*

**DOI:** 10.3390/antibiotics11010030

**Published:** 2021-12-28

**Authors:** Yi Yuan, Jesus D. Rosado-Lugo, Yongzheng Zhang, Pratik Datta, Yangsheng Sun, Yanlu Cao, Anamika Banerjee, Ajit K. Parhi

**Affiliations:** TAXIS Pharmaceuticals, Inc., 9 Deer Park Drive, Suite J-15, Monmouth Junction, NJ 08852, USA; yyuan@taxispharma.com (Y.Y.); jrosado@taxispharma.com (J.D.R.-L.); yzhang@taxispharma.com (Y.Z.); pdatta@taxispharma.com (P.D.); yangsheng.sun@gmail.com (Y.S.); yanlucao@gmail.com (Y.C.); anamikabanerjee375@gmail.com (A.B.)

**Keywords:** *Pseudomonas aeruginosa*, RND efflux pumps, efflux pump inhibitors, heterocyclic carboxamides, antimicrobial drug resistance

## Abstract

The ability to rescue the activity of antimicrobials that are no longer effective against bacterial pathogens such as *Pseudomonas aeruginosa* is an attractive strategy to combat antimicrobial drug resistance. Herein, novel efflux pump inhibitors (EPIs) demonstrating strong potentiation in combination with levofloxacin against wild-type *P. aeruginosa* ATCC 27853 are presented. A structure activity relationship of aryl substituted heterocyclic carboxamides containing a pentane diamine side chain is described. Out of several classes of fused heterocyclic carboxamides, aryl indole carboxamide compound **6j** (TXA01182) at 6.25 µg/mL showed 8-fold potentiation of levofloxacin. TXA01182 was found to have equally synergistic activities with other antimicrobial classes (monobactam, fluoroquinolones, sulfonamide and tetracyclines) against *P. aeruginosa*. Several biophysical and genetic studies rule out membrane disruption and support efflux inhibition as the mechanism of action (MOA) of TXA01182. TXA01182 was determined to lower the frequency of resistance (FoR) of the partner antimicrobials and enhance the killing kinetics of levofloxacin. Furthermore, TXA01182 demonstrated a synergistic effect with levofloxacin against several multidrug resistant *P. aeruginosa* clinical isolates.

## 1. Introduction

Efflux is one of the major resistance mechanisms in Gram-negative bacteria [1,2]. Efflux pumps are transporter proteins involved in the extrusion of a wide range of antimicrobials from the bacterial cell [3]. Many of these efflux pumps in Gram-negative bacteria belong to the resistance-nodulation-cell division (RND) family of tripartite efflux pumps [4,5]. The RND efflux systems in *Pseudomonas aeruginosa* consist of many identified efflux pumps that are largely responsible for its multidrug resistance [6]. Four well characterized multidrug efflux pump systems (MexA-MexB-OprM, MexC-MexD-OprJ, MexE-MexF-OprN, and MexX-MexY-OprM) are prevalent in multiple clinical isolates of *P. aeruginosa* [7]. In a recent study it was reported that in 30 clinical strains of *P. aeruginosa,* overexpression of *mexB*, *mexF* and *mexY* was detected in 27, 12, and 45% of the clinical strains, respectively [8]. Additionally, *P. aeruginosa* possesses six RND efflux pumps (MexJK, MexGHI-OpmD, MexVW, MexPQ-OpmE, MexMN, and TriABC-OpmH) which might contribute to resistance at the clinic [9]. Approaches for circumventing efflux-mediated resistance require the development of direct, as well as indirect, antimicrobial agents. New antibiotics that are poor substrates of efflux pumps can serve as direct antimicrobial agents. On the other hand, indirect antimicrobial agents involve novel drugs that need to be paired with antimicrobials that have been rendered inactive by efflux. Due to dwindling pipelines for new antibiotics, inhibition of efflux pumps has become an attractive avenue to rejuvenate older antimicrobials to tackle the antibiotics resistance problem [10,11]. An ideal efflux pump inhibitor (EPI) would efficiently block these pumps, increasing the concentrations of the antimicrobials within the cell, thus rendering them effective again [12].

A variety of chemical scaffolds have been shown to act as EPIs (Figure 1). Among the first EPIs is the peptidomimetic MC-207,110 (Phenylalanine-arginine β-naphthylamide (PAβN)), and related dipeptide amide compound, MC-04,124. Both were developed by Essential Therapeutics Inc and were shown to potentiate levofloxacin effectively in wild-type and efflux-pump-overexpressed *P. aeruginosa* strains [13,14]. A non-peptidomimetic lead series worthy of mentioning is D13-9001, the lead candidate from a novel pyridopyrimidine scaffold-based series developed for MexAB-OprM specific RND-pump inhibition [15]. A more recent endeavor involves MBX-2319, from a pyrazolopyridine class of compounds that potentiates ciprofloxacin, levofloxacin, and piperacillin against *E. coli* [16]. However, the development of these compounds has stopped or stalled for various reasons. PAβN and related compounds showed prolonged accumulation in tissues associated with renal toxicity [17]. MC-04,124 was discontinued following the closure of Essential Therapeutics [18]. The MBX EPI series does not inhibit efflux in *P. aeruginosa* and, thus, may not be suitable for anti-pseudomonal therapy [19]. Despite presenting good in vitro activity and being efficacious in vivo against *P. aeruginosa*, D13-9001 is yet to progress to clinical evaluation [20]. Thus, the discovery of novel EPI scaffolds active against Gram-negative pathogens, particularly *P. aeruginosa*, with the potential of clinical use is still needed.

While no EPI so far has been approved for clinical use due to various reasons [17,21], the need to combat antimicrobial resistant Gram-negative pathogens, especially ESKAPE pathogens (*Enterococcus faecium*, *Staphylococcus aureus*, *Klebsiella pneumoniae*, *Acinetobacter baumannii*, *P. aeruginosa*, and *Enterobacter* species), is of urgent medical need [22]. TAXIS Pharmaceuticals is committed [23] to finding solutions to multidrug resistant bacterial infections, and has recently published preliminary efforts on a diaminopentanamide class of potentiators (Figure 2) [24,25]. These potentiators had a hydrophobic arylalkyl or hetero aryl alkyl head groups linked by either an amide (compound **1**) or by a reversed amide (compound **2**) to chiral diamines. Interestingly, while amide **1** enhances the activity of the macrolide clarithromycin by 4-fold against *E. coli* ATCC 25922, the reversed amide **2** showed 16-fold potentiation at 12.5 µg/mL against *E. coli* ATCC 25922 with clarithromycin. Cell-based membrane permeabilization assays have determined an undesired mixed mode of action with these initial hits. Additionally, cationic amphiphilic molecules from this series were deemed too risky for further development, as they are prone to intracellular accumulation. The present study aims to find a development-worthy EPI series with better solubility, low lipophilicity and appreciable microsomal stability, which will serve as a starting point for a more elaborate optimization process. Lessons learned from the failure of earlier programs have been considered carefully as TAXIS continues to reach its goal of a first-in-class EPI-antibiotic combination treatment option against *P. aeruginosa* infections.

This report presents the results of a synthetic screen seeking to replace the aryl alkyl head groups by more druggable fragments connected to the chiral triamine **5** side chain of compounds **1** and **2** [24,25] (Figure 3). A key design aspect of this effort was to eliminate membrane disruption, a MOA seen in other EPI programs. It is anticipated that toxicity due to nonspecific binding may be eliminated or reduced with efflux specific mechanisms. Another aim of this series is to improve the metabolic and serum stability of the EPI molecules as compared to the peptide or peptidomimetic EPIs, which may benefit their potential development in the future. This work produced TXA01182, a novel EPI that enhances the activity of multiple classes of antimicrobials with efflux liabilities in wild-type and multidrug-resistant clinical isolates of *P. aeruginosa*, while playing a minimal role in membrane disruption and avoiding the onset of resistance.

## 2. Results and Discussion

Efflux pumps play a prominent role in the multidrug resistance of *P. aeruginosa* and many other Gram-negative bacteria. These efflux pumps have different substrate specificities, and their production and activity can be increased by many factors commonly present in infections. Moreover, many commonly used antibiotics can select mutants that constitutively overproduce efflux pump systems in *P. aeruginosa* (e.g., fluoroquinolones and aminoglycosides induce the overexpression of MexCD-OprJ and MexXY-OprM, respectively) [26,27]. As such, a standalone antibiotic would face a profound risk of being rendered ineffective almost immediately. Recognizing that efflux pumps contribute significantly to mediating antibiotic resistance in Gram-negative bacteria, TAXIS Pharmaceuticals has focused on the inhibition of these pumps as a first-in-class treatment option where resistance is an issue. Such use of EPIs obviates the need to discover new antibiotics, a strategy that saves a lot of time, effort, and capital associated with discovery and development of antibiotics. Furthermore, this allows clinicians to exploit the already well-established pharmacological properties of known antibiotics. A particularly important implication of EPIs as therapeutic agents is their ability to overcome resistance, thereby enabling the use of already optimized and stockpiled antibiotics.

### 2.1. Screening Results

The structures of the synthesized heterocyclic carboxamides **6a**–**n** that are screened for this study are shown in Table 1.

The carboxamide hydrochloride salts (**6a** to **6n**) were first assayed for their antibacterial activities against *P. aeruginosa* ATCC 27853 (Table 2). Except for the two benzothiophene analogs **6g** and **6h** with MICs at 25 µg/mL, all the other tested compounds had MICs ≥ 100 µg/mL. The ability of heterocyclic carboxamide hydrochloride salts (**6a** to **6n**) to lower the MIC of levofloxacin (LVX) was tested. Most compounds potentiated LVX in a concentration-dependent manner (data not shown). The lowest active concentration for the compounds was 6.25 µg/mL (Table 2). Thus, the 6.25 µg/mL concentration was chosen for antimicrobial potentiation assays to ensure that the MICs of the screened compounds did not interfere with their potentiation properties. LVX, as the choice of antimicrobial for the screens, stems from the fact that the *P. aeruginosa* efflux pumps MexAB-OprM, MexCD-OprJ, MexEF-OprN and MexXY-OprM play an important role in susceptibility to fluoroquinolones in vitro [28,29,30].

The fused heterocyclic carboxamides were separated into two groups: (1) from zero to weak potentiation (benzothiazole **6a** and **6b**, and benzimidazole **6c** and **6d**) and (2) from moderate to high potentiation (benzothiophene **6g** and **6h**, indole **6i** to **6l**). Although benzothiophene compound **6g** showed the highest potency (32-fold at 6.25 µg/mL), due to its low MIC, this scaffold was discarded from further screening to maintain the strict guideline of at least a 12-fold difference in MIC and the screening concentration. The 6-substituted *para*-fluorophenyl showed better activity than substitution at 5-position of indoles (**6j** vs. **6i**; **6l** vs. **6k**). Azaindole analogs **6m** and **6n** completely failed to exhibit potentiation as compared to the matched carbon analogs **6i** and **6j**. The potentiation of LVX by 3 pairs of enantiomers (**6c** and **6d**; **6i** and **6k**; **6j** and **6l**) was investigated, while the *S*-enantiomer **6i** was 2-fold better than *R*-enantiomer **6k**; the other two pairs of EPI enantiomers did not show any differences in their potentiation effects. The *S*-enantiomer **6j** was selected as the model compound for further study due to the easier access to l-ornithine than d-ornithine as the starting material for the synthesis of the diamine side chain. It should be pointed out that while the configuration of the chiral amine did not affect the potentiation significantly in this case, the metabolic properties (e.g., metabolic clearance and PK) might vary between the stereoisomers. Screening the active compounds in *P. aeruginosa* mutants overexpressing MexAB-OprM, MexCD-OprJ, MexEF-OprN and MexXY-OprM did not lead to a significant increase in the potentiation, as seen with the ATCC 27853 strain (Appendix A). This interesting result might suggest that if the compounds in Table 2 are inhibiting efflux, they might be doing so by acting on a different efflux pump.

### 2.2. In-Depth Potentiation Evaluation of **6j** (TXA01182)

Based on the results of the previous screen, one of the most potent compounds **6j**, TXA01182, was further studied at 6.25 µg/mL in combination with different classes of antimicrobials in *P. aeruginosa* ATCC 27853 (Table 3).

Since the initial screening platform utilized levofloxacin, it was not surprising that other fluoroquinolones (ciprofloxacin and moxifloxacin) were also potentiated by TXA01182. The low potentiation (2-fold) with ciprofloxacin could be attributed to its low MIC, or to it being effluxed by a pump not targeted by TXA01182 potentially. TXA01182′s effect on tetracycline antimicrobials (doxycycline, minocycline and tigecycline) was pronounced with minocycline, showing 32-fold potentiation. Potentiation of tetracyclines seems to correlate with the efflux liability of these antimicrobials, as minocycline and doxycycline are better substrates to efflux pumps than tigecycline [32,33,34]. It is noteworthy that TXA01182 potentiates tigecycline 4-fold, bringing its MIC back to a clinically relevant level. As both cotrimoxazole and chloramphenicol fail to show MICs against wild-type *P. aeruginosa*, the ability to see any MIC (16 and 32 µg/mL, respectively) in the presence of TXA01182 was encouraging, and in accordance with both antimicrobials being substrates of efflux pumps [35,36]. As anticipated for an EPI, imipenem and gentamicin, which are not the substrates of RND efflux pumps in *P. aeruginosa*, were not potentiated by TXA01182 [37]. The combination with aztreonam, azithromycin, cefepime and ceftazidime with TXA01182 demonstrated only minimal potentiation (2- to 4-fold), suggesting that TXA01182 does not inhibit the pumps responsible for the efflux of these antimicrobials.

### 2.3. TXA01182 Plays a Minimal Role in Membrane Disruption

To test if the synthetic efforts presented in Table 1 eliminated membrane disruption as a MOA, the possibility of membrane disruption caused by TXA01182 was examined. Membrane disruption was assayed with two approaches: (1) a flow cytometry-based propidium iodide (PI) assay to monitor inner membrane permeabilization, and (2) a nitrocefin (NCF) assay to monitor outer membrane permeabilization. NCF is a chromogenic cephalosporin that changes from yellow to red when the amide bond in the β-lactam ring is hydrolyzed by a β-lactamase. The rate of hydrolysis in intact cells is slow as it is limited by the rate of diffusion of periplasmic β-lactamase across the outer membrane. However, in the presence of an agent that permeabilizes the outer membrane, the rate of hydrolysis will increase [38]. The impact of TXA01182 on NCF hydrolysis is minimal at concentrations below 12.5 μg/mL, suggesting that TXA01182 does not interact with the outer membrane of *P. aeruginosa* at these concentrations (Figure 4A). This is a clear improvement from TAXIS’ previous scaffolds, which showed clear outer membrane disruption at 3.13 μg/mL [24]. Polymyxin B was used as a positive control (Figure 4B). TXA01182 also did not disrupt the bacterial inner membrane below concentrations of 25 μg/mL compared with DMSO and polymyxin B as vehicle and positive controls, respectively (Figure 4C). In the PI assay, log-phase *P. aeruginosa* cells were mixed with various concentrations of TXA01182, followed by the addition of PI. Cells whose membranes remain intact exclude PI and remain non-fluorescent, while cells with compromised membrane integrity allow PI to enter and bind to DNA, resulting in fluorescence.

### 2.4. TXA01182 Inhibits the Efflux of Ethidium Bromide

The efflux of ethidium bromide (EtBr) by *P. aeruginosa* cells in the presence of TXA01182 was studied in order to establish efflux inhibition as the MOA (Figure 5). Wild- type *P. aeruginosa* PAO1 cells were incubated with EtBr to allow for intracellular accumulation and treated with CCCP to inhibit active efflux. When bound to intracellular bacterial DNA, EtBr fluoresces brightly, while any unbound EtBr outside bacterial cells exhibit little or no fluorescence. Following activation by the addition of glucose, the efflux of EtBr can be followed in real time as a decrease in fluorescence based on the concentration of TXA01182. As seen in Figure 5A, the fluorescence intensity increased proportionally with increasing concentration of TXA01182, indicating intracellular accumulation of EtBr and supporting a role in efflux inhibition by TXA01182. In contrast, polymyxin B, a known membrane disruptor, did not lead to intracellular accumulation of EtBr, in line with its MOA (Figure 5B).

### 2.5. TXA01182 Enhances the Activity of Levofloxacin against Clinical Isolates of P. aeruginosa

Eight multidrug-resistant (MDR) *P. aeruginosa* strains from the CDC and FDA antibiotic resistance isolate bank resistant to levofloxacin (MICs ranging from 8 to 64 μg/mL) were used to test the EPI activity of TXA01182. The resistance determinants of these strains, including those associated with levofloxacin resistance and efflux upregulation, are shown in Table 4. The *gyrA-T83I or gyrA-T133H* mutations lead to fluoroquinolone resistance [39,40,41], while the *nalC-G71E* and *mexR-V126Q* mutations are associated with MexAB-OprM overexpression [42,43,44,45,46]. TXA01182 potentiated levofloxacin in all eight resistant isolates from 8- to 32-fold at 6.25 μg/mL. It also potentiated ciprofloxacin and tigecycline from 2- to 16-fold at the same concentration (Appendix A). In contrast, MC-04,124, PAβN and CCCP, three well characterized EPIs mostly failed to show any potentiation. PAβN was able to enhance the activity of levofloxacin 8-fold on two out of six strains (AR-0239 and AR-0249) but at a higher EPI concentration of 50 μg/mL. It is noteworthy that the TXA01182 and levofloxacin combination was beneficial against these strains, irrespective of their resistance determinants. The ability of TXA01182 to lower levofloxacin’s MIC in these eight resistant clinical isolates suggests that the TXA01182 class of EPI combined with appropriate antimicrobials may be helpful in the fight against multidrug resistant bacteria in the clinic.

### 2.6. TXA01182 Lowers the Frequency of Resistance to Levofloxacin

In addition to reducing the levels of intrinsic resistance, a potent EPI is also expected to significantly reverse acquired resistance as well as decrease the frequency at which antimicrobial resistance emerges. To test whether TXA01182 in combination with antimicrobials increases the selective pressure on resistant mutant emergence, the effect of TXA01182 on the frequency of resistance (FoR) of *P. aeruginosa* ATCC 27853 to levofloxacin was tested. TXA01182 reduced the FoR to levofloxacin approximately 500-fold (Table 5). The FoR to TXA01182 alone was close to 1, indicating that it did not affect the growth of *P. aeruginosa*. TXA01182 also reduced the FoR to cefepime 10-fold (Appendix A). The undetectable levels of resistance seen in the levofloxacin-TXA01182 combination would be of great value in a clinical setting, particularly in cystic fibrosis patients infected with *P. aeruginosa,* where patients are colonized by hypermutable strains that persist for years [47].

### 2.7. Time-Kill Assay

In addition to assessing the potentiation activity of TXA01182 in vitro, its potentiation of a minimally bactericidal concentration of levofloxacin (1X MIC) was probed against *P. aeruginosa* ATCC 27853 with time-kill studies. Figure 6 shows time-kill curves with levofloxacin in the absence or presence of TXA01182. By itself, TXA01182 had no effect on the growth of *P. aeruginosa* ATCC 27853 at the highest concentration tested (gray curve). TXA01182 enhanced levofloxacin’s killing kinetics in a concentration dependent manner (green and red curves). After 3 h of incubation, the combination of levofloxacin and TXA01182 killed more bacteria by a magnitude of >2-logs compared to levofloxacin alone (orange curve). After 6 h of incubation, the combination of levofloxacin and TXA01182 killed more bacteria by a magnitude of >3-logs compared to levofloxacin alone. At 24 h, the combination of levofloxacin and TXA01182 achieved almost 6-logs of kill more than levofloxacin alone (orange vs. red curves). These results suggest that the killing kinetics for the combination of levofloxacin and TXA01182 are faster than those of levofloxacin alone. The 24 h period regrowth associated with antimicrobial resistance was also greatly diminished by the levofloxacin-TXA01182 combination. The potentiation of a minimally bactericidal concentration of levofloxacin (1X MIC) by TXA01182 seen in Figure 6 is consistent with similar studies carried out with a ciprofloxacin-MBX-2319 combination in *E. coli* [15].

## 3. Materials and Methods

### 3.1. Synthesis

The general synthetic scheme of heterocyclic carboxamides **6a**–**n** is outlined in Figure 7. For screening purposes, the chiral center of di-Boc protected pentane diamine was fixed as either *R-* or *S-* configuration. SAR found the aryl-linked heterocycles to be superior to the original di-aryl leads (Compounds **1** and **2**). It is noteworthy to mention that without any aryl substitutions on the fused heterocycles, these compounds were typically devoid of any potentiation activity (data not shown). To improve metabolic stability, *para-*fluoro substituted aryl boronic acids were typically chosen. Coupling with commercially available or known 5 or 6- bromo-biarylheterocyclic-2-carboxylic methyl or ethyl esters **3** with *p*-fluorophenyl boronic acid under Suzuki reaction conditions afforded the desired esters **4a**–**n** in high yields. Hydrolysis of the esters to the corresponding heteroaryl acids followed by coupling with the chiral di-Boc amine **5** produced the protected carboxamides. The final compounds were deprotected with HCl in dioxane to afford the dihydrochloride salts of heterocyclic carboxamides **6a**–**n** (Table 1) in excellent yields. The structures of the intermediates and the final compounds were analyzed by ^1^H-NMR spectra and LC/MS data. As a representative example, the experimental procedure and spectral data of **6j** (TXA01182) are provided (Figure 8). The spectral data of other compounds can be found in the Appendix A.

Step (1) Synthesis of methyl 6-bromo-1*H*-indole-2-carboxylate **3j**

To a suspension of 6-bromo-1*H*-indole-2-carboxylic acid **7** (5.0 g, 20.8 mmol) in MeOH (100 mL) was added SOCl_2_ (2.26 mL, 31 mmol) very slowly. The mixture was heated under reflux until TLC showed no starting material left. Solvent was removed under vacuo and the crude product was collected as a brown powder (5.2 g, 98% yield) after drying. It was used for next step reaction without purification. ^1^H NMR (300 MHz, CDCl_3_) δ 8.88 (s br, 1H), 7.59 (s, 1H), 7.55 (d, *J* = 6 Hz, 1H), 7.25 (m, 1H), 7.19 (s, 1H), 3.96 (s, 3H).

Step (2) Synthesis of methyl 6-(4-fluorophenyl)-1*H*-indole-2-carboxylate **9**

The mixture of methyl 6-bromo-1*H*-indole-2-carboxylate **3****j** (510 mg, 2 mmol), (4-fluorophenyl)boronic acid **8** (520 mg, 4 mmol) in a mixture of toluene, methanol and saturated Na_2_CO_3_ solution (18/4/4 mL) was degassed and Pd(dppf)Cl_2_ (60 mg, 0.08 mmol) was added. The reaction mixture was heated at 100 °C for 3 hrs and it was extracted with EtOAc and washed with brine and concentrated. Then, it was purified by column chromatography on silica gel (10–30% ethyl acetate/hexanes) to give the product (390 mg, 72% yield) as an off-white powder. ^1^H NMR (300 MHz, CDCl_3_) δ 8.94 (s br, 1H), 7.74 (d, *J* = 8.4 Hz, 1H), 7.62 (m, 3H), 7.36 (d, *J* = 8.4 Hz, 1H), 7.24 (m, 1H), 7.15 (m, 2H), 3.96 (s 3H).

Step (3) Synthesis of 6-(4-fluorophenyl)-1*H*-indole-2-carboxylic acid **4j**

To a solution of methyl 6-(4-fluorophenyl)-1*H*-indole-2-carboxylate **9** (0.35 g, 1.3 mmol) in THF was added NaOH solution (2 M, 5 mL). It was stirred at room temperature until no starting material was left. THF was removed under vacuo and the residue was acidified with an HCl solution. The precipitate was filtered and washed with water. It was dried to provide the product as an off-white powder (290 mg, 88% yield) which was used for next step reaction without further purification. ^1^H NMR (300 MHz, CDCl_3_) δ 11.48 (s br, 1H), 7.68 (s, 1H), 7.65 (m, 1H), 7.64 (m, 1H), 7.60 (m, 1H), 7.55 (d, *J* = 8.4 Hz, 1H), 7.26 (m, 1H), 7.19 (m, 1H), 6.66 (s, 1H). MS (ESI−) *m*/*z*: calcd for C_15_H_9_FNO_2_. [M − H]^−^: 254.06; found 254.05.

Step (4) Synthesis of di-*tert*-butyl (5-(6-(4-fluorophenyl)-1*H*-indole-2-carboxamido)pentane-1,4-diyl)(*S*)-dicarbamate **10**

To a solution of 6-(4-fluorophenyl)-1*H*-indole-2-carboxylic acid **4j** (60 mg, 0.24 mmol) in anhydrous DMF (2 mL) was added DIPEA (0.09 mL, 0.5 mmol), HOBt (22 mg, 0.18 mmol) and EDC (45 mg, 0.24 mmol). The reaction mixture was stirred at room temperature and di-*tert*-butyl (5-aminopentane-1,4-diyl)(*S*)-dicarbamate **5** [24] (76 mg, 0.24 mmol) was added. Stirring of the reaction mixture was continued at room temperature overnight. Then, it was diluted with EtOAc and washed with water and brine. The organic layer was dried over anhydrous sodium sulfate and filtered. The filtrate was concentrated and purified by column chromatography on silica gel (40–60% ethyl acetate/hexanes) to give the product (75 mg, 57% yield) as a white solid. ^1^H NMR (300 MHz, CDCl_3_) δ 7.57 (m, 1H), 7.48 (m, 2H), 7.33 (m, 1H), 7.18 (d, *J* = 8.1 Hz, 1H), 6.99 (t, *J* = 8.1 Hz, 2H), 6.92 (s, 1H), 5.78 (br, 1H), 3.57 (m, 1H), 3.30 (m, 2H), 2.95 (m, 2H), 1.43 (m, 4H), 1.28 (s, 9H), 1.24 (s, 9H).

Step (5) Synthesis of (*S*)-*N-*(2,5-diaminopentyl)-6-(4-fluorophenyl)-1*H*-indole-2-carboxamide dihydrochloride salt **6j** (TXA01182)

To a solution of di-*tert*-butyl (5-(6-(4-fluorophenyl)-1*H*-indole-2-carboxamido)pentane-1,4-diyl)(*S*)-dicarbamate **10** (25 mg, 0.045 mmol) in MeOH (10 mL) HCl solution (4 M in dioxane, 0.2 mL) was added. It was stirred at room temperature overnight and solvent was removed under vacuo. The residue was triturated with EtOAc and the precipitate was collected as an off-white powder (15 mg, 70% yield). ^1^H NMR (300 MHz, CD_3_OD) δ 7.69 (d, *J* = 8.4 Hz, 1H), 7.67 (m, 1H), 7.66 (m, 2H), 7.34 (dd, *J* = 1.8, 8.4 Hz, 1H), 7.21 (s, 1H), 7.16 (t, *J* = 8.7 Hz, 2H), 3.76 (m, 1H), 3.62 (m, 1H), 3.50 (m, 1H), 3.02 (m, 2H), 1.87 (m, 4H). ^13^C NMR (75 MHz, CD_3_OD) δ 163.87, 160.65, 138.10, 137.60, 136.58, 130.76, 128.53, 128.43, 126.84, 121.94, 119.71, 115.13, 114.84, 109.79, 104.21, 51.85, 40.60, 38.91, 27.15, 23.23. MS (ESI+) *m*/*z*: calcd for C_20_H_24_FN_4_O [M + H]^+^: 355.19; found 355.20.

### 3.2. Bacterial Strains, Media, and Reagents

*P. aeruginosa* ATCC 27853 was obtained from the American Type Culture Collection (ATCC). *P. aeruginosa* multidrug-resistant isolates AR-0229, AR-0232, AR-0234, AR-0239, AR-0244, AR-0246, AR-0249, AR-0264 were obtained from the CDC and FDA Antibiotic Resistance Isolate Bank. Wild-type *P. aeruginosa* PAO1 is a common laboratory strain. *P. aeruginosa* strains K767 (WT), K1455 (*mexAB-oprM* overexpressed), K2415 (*mexXY-oprM* overexpressed), K2951 (*mexCD-oprJ* overexpressed) and K2376 (*mexEF-oprN* overexpressed) were obtained from Prof. Keith Poole (Queen’s University, Kingston, ON, Canada) and have been characterized elsewhere [48,49,50,51]. Bacterial cells were grown in cation-adjusted Mueller Hinton (CAMH) media, brain heart infusion broth (BHI) or tryptic soy agar (TSA) plates all obtained from Becton, Dickinson, and Company (BD, Franklin Lakes, NJ, USA). Aztreonam, cefepime, ceftazidime, ciprofloxacin, moxifloxacin, levofloxacin, minocycline, tigecycline, chloramphenicol and imipenem were purchased from TOKU-E (Bellingham, WA, USA). Azithromycin was purchased from Tokyo Chemical Industry (Portland, OR, USA). Cotrimoxazole was purchased from Toronto Research Chemicals (North York, ON, Canada). Doxycycline, gentamicin, phenylalanine-arginine β-naphthylamide (PAβN) and polymyxin B were purchased from Sigma-Aldrich (St. Louis, MO, USA). Ethidium bromide (EtBr) and glucose were purchased from VWR (Radnor, PA, USA). MC-04,124 was synthesized at TAXIS Pharmaceuticals. Carbonyl cyanide 3-chlorophenylhydrazone (CCCP) was purchased from Enzo Life Sciences (Farmingdale, NY, USA).

### 3.3. Minimum Inhibitory Concentration (MIC) Assay for Potentiation of Antimicrobial Activity against P. aeruginosa

An MIC-based assay was used to evaluate the ability of all tested compounds to act as potentiators of antimicrobial activity against *P. aeruginosa* ATCC 27853. MICs were conducted using a protocol similar to that described previously [24]. Log-phase bacteria were added to 96-well microtiter plates (at 5 × 10^5^ colony forming units (CFU) per mL) containing two-fold serial dilutions of the antimicrobial in CAMH broth, either in the absence or presence of each test compound **6a**–**n** (at a concentration of 6.25 μg/mL). The final volume in each well was 0.1 mL, and the microtiter plates were incubated aerobically for 18–24 h at 37 °C. Bacterial growth was then monitored by measuring the optical density (OD) at 600 nm using a VersaMax^®^ plate reader (Molecular Devices, Inc., Sunnyvale, CA, USA), with the MIC being defined as the lowest antimicrobial concentration at which bacterial growth was inhibited compared to antimicrobial-free and compound-free controls. The MIC of each test compound against *P. aeruginosa* ATCC 27853 was also determined using a similar assay, with the exception that the microtiter plates contained serial dilutions of test compound rather than antimicrobial.

### 3.4. Fluorescence-Activated Cell Sorting (FACS) Assay for Permeabilization of the Outer and Inner Cell Membranes to Propidium Iodide (PI) in P. aeruginosa

The FACS assay used for assessing the potential of test compounds to permeabilize both the outer and inner cell membranes of *P. aeruginosa* bacterial cells to PI was conducted as described previously [24]. Log-phase *P. aeruginosa* ATCC 27853 bacterial cells grown in CAMH broth were diluted in PBS to a concentration of 4 × 10^6^ CFU/mL. The bacteria were aliquoted into tubes and mixed with TXA01182 at concentrations ranging from 1/4th to 1/64th times the MIC (25 to 1.56 μg/mL). DMSO alone was used as a solvent control. Polymyxin B was used as a positive control. All test concentrations and controls were prepared in triplicate. PI was then added to all tubes at a final concentration of 50 μM, and the samples were incubated in the dark at room temperature for 1 h. Intracellular PI fluorescence was detected by flow cytometry using a Gallios Cytometer (Beckman Coulter Inc., Indianapolis, IN, USA). The 488 nm laser was used for excitation, with the 620/630 nm channel being used for emission. For each sample, the fluorescence of 30,000 individual bacterial cells was measured, and the percent of cells that stained positive for PI fluorescence was calculated.

### 3.5. Nitrocefin (NCF) Cellular Assay for Outer Cell Membrane Permeabilization Assessment in P. aeruginosa

*P. aeruginosa* ATCC 27853 was incubated with imipenem 1/4th MIC overnight to induce the expression of chromosomal β-lactamase in CAMH media. Cells were harvested, washed in 10 mM HEPES, 2.5 mM MgCl_2_, pH = 7 buffer and resuspended in the same buffer at an OD_600_ of approximately 0.5. Then, 100 μL of the cell suspension were mixed with 50 μL of either TXA01182 or polymyxin B to give a final concentration of 0 to 25 μg/mL, or 0 to 8 μg/mL, respectively. Next, 50 μL of nitrocefin was added to reach a final concentration of 32 μg/mL. Hydrolysis of nitrocefin was monitored spectrophotometrically by measurement of the increase in absorbance at 490 nm. Assays were performed in 96-well plates in a SpectraMax iD5 spectrophotometer (Molecular Devices, Sunnyvale, CA, USA).

### 3.6. Fluorescence-Based Cellular Assay for Inhibition of Pump-Mediated Efflux of Ethidium Bromide (EtBr)

The potential of TXA01182 to inhibit efflux pumps in *P. aeruginosa* was evaluated using a fluorescence-based cellular assay that measures the pump-mediated efflux of EtBr [24,52]. In this assay, *P. aeruginosa* PAO1 cells were harvested from overnight cultures by centrifugation, and the cell pellets were washed with PBS containing 1 mM MgCl_2_ (PBSM). After washing the cells, the cell pellets were resuspended in PBSM to achieve a final OD_600_ of 1.0. The proton gradient required for RND efflux pumps to function was then uncoupled by addition of carbonyl cyanide 3-chlorophenylhydrazone (CCCP) to a final concentration of 50 μM, along with the addition of EtBr at a final concentration of 200 µM. Cells were then incubated in the dark at 37 °C for 50 min to allow for EtBr to accumulate inside cells. Then, 200 μL aliquots of the bacterial suspension were distributed into wells of a black, flat-bottom 96-well plate containing TXA01182 at concentrations ranging from 1/4th to 1/32nd times the MIC (25 to 3.13 μg/mL), or an equivalent volume of the vehicle (DMSO) alone. A plate vortexer was used to mix the bacterial cells with the test compounds. After pre-incubation at 37 °C for 5 min, efflux pump activity was initiated by reenergizing the bacterial cells with the addition of glucose to a final concentration of 100 mM. EtBr efflux was monitored using a SpectraMax^®^ 2 fluorescent plate reader (Molecular Devices, Inc., Sunnyvale, CA, USA) to measure the fluorescence of each well at 37 °C once per minute for 240 min. The excitation and emission wavelengths were set at 510 and 620 nm, respectively.

### 3.7. Frequency of Resistance (FoR) Studies

The FoR of *P. aeruginosa* to levofloxacin or cefepime alone, or in combination with TXA01182, was assayed by using a large-inoculum approach described previously [53]. Tryptic soy agar plates were prepared containing 4 μg/mL levofloxacin alone (four times the MIC), 4 μg/mL levofloxacin plus 6.25 μg/mL TXA01182, 2 μg/mL cefepime alone (two times the MIC), 2 μg/mL cefepime plus 6.25 μg/mL TXA01182, or DMSO alone. All plates were incubated at 37 °C and examined after 72 h. FoR was calculated by dividing the number of resistant colonies growing on antimicrobial-containing plates, or antimicrobial plus TXA01182-containing plates by the total number of CFU in the initial test inoculum.

### 3.8. Time-Kill Studies

Time–kill studies were performed according to previously published methods [54]. Briefly, freshly prepared colonies were resuspended in 5 mL of BHI and incubated with shaking (37 °C, 180 rpm) overnight. Cultures were then diluted to 5 × 10^6^ CFU/mL. When indicated, levofloxacin was added to the prepared bacterial suspensions at one time the MIC (1 μg/mL). TXA01182 was added to bacterial suspensions at 1/16th, 1/25th and 1/32nd times the MIC (6.25 μg/mL, 4 μg/mL, and 3.125 μg/mL, respectively). A growth control with only media was included. Controls with only levofloxacin or TXA01182 were also included. The starting inoculum was determined from the growth control tube immediately after dilution and was recorded as the count at time zero. Cultures were incubated with shaking (37 °C, 180 rpm), and viability counts were performed at 3, 6, and 24 h by plating serial dilutions on tryptic soy agar (TSA). TSA plates were incubated at 37 °C for at least 18 h. Colonies were counted, and the results were recorded as the number of CFU/mL.

## 4. Conclusions

As global resistance to conventional antibiotics rises, in part, by efflux-mediated mechanisms, new strategies are needed to develop future novel therapeutics. A promising strategy for combating efflux-mediated resistance is the combination of antimicrobials with indirect antimicrobial agents such as EPIs. Thus, there is an urgent need for effective efflux inhibitors that can restore the activity of conventional antibiotics. The aim of this program was to discover novel EPIs active against *P. aeruginosa* within a heterocyclic-carboxamide series. A variety of fused heterocyclic carboxamides were synthesized and screened for their potentiation effect on levofloxacin against *P. aeruginosa* ATCC 27853. The 2-carboxamide indole compound **6j** (TXA01182) potentiated levofloxacin prominently, while other heterocyclic carboxamides were either inactive or had a low potentiation effect. Based on the good potentiation ability of TXA01182 on levofloxacin, it was further evaluated in combination with different classes of antimicrobials. The MOA of this hit compound was studied biophysically to prove that it acts as an EPI in *P. aeruginosa*. Inhibition by TXA01182 of an efflux pump yet to be identified appears to be of clinical relevance, since the susceptibility of MDR *P. aeruginosa* clinical isolates to levofloxacin and tigecycline is increased significantly by TXA01182. Identification of the efflux pump(s) targeted by TXA01182, and its analogs, is still ongoing. In terms of druggable properties, indole-containing EPIs similar to TXA01182, which are currently under development, are particularly promising on the basis of their lower lipophilicity, excellent solubility, and high metabolic stability (data not shown). Attenuating the pKa of both the amines while retaining the potentiation effects of these EPIs might reduce the risks of any amine induced toxicity, including cardiotoxicity and nephrotoxicity, during developmental phases. The detailed SAR studies and optimization efforts within this class of EPIs will be reported after this publication. The results of the studies, with selected optimized candidates involving broad-spectrum potentiation abilities, detailed MOA, improvement of toxicity profile, including hERG binding by structure property relationship, PK parameters and, lastly, in vivo efficacy, and will be published in due course. 

## 5. Patents

Some works of this study are published in the following patent applications: LaVoie, E.; Parhi, A.; Yuan, Y.; Zhang, Y.; Sun, Y. Indole Derivatives as Efflux Pump Inhibitors. WO2018165611. LaVoie, E.; Parhi, A.; Zhang, Y.; Yuan, Y.; Sun, Y. Bacterial Efflux Pump Inhibitors. WO2018165612.

## Figures and Tables

**Figure 1 antibiotics-11-00030-f001:**
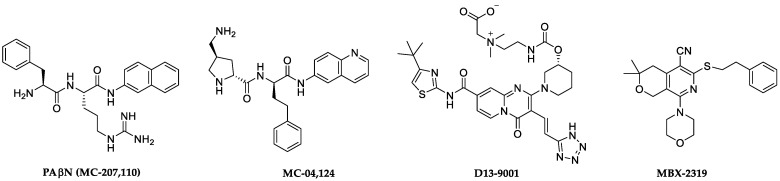
Chemical structures of representative lead efflux pump inhibitors.

**Figure 2 antibiotics-11-00030-f002:**
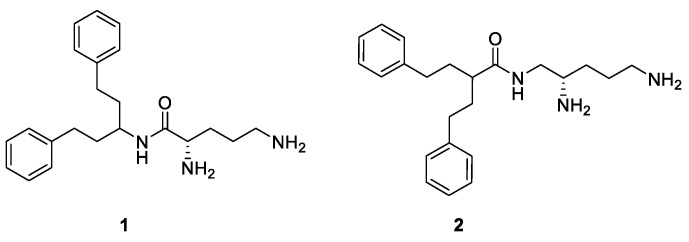
Structures of the two-aryl alkyl diaminopentanamide potentiators.

**Figure 3 antibiotics-11-00030-f003:**
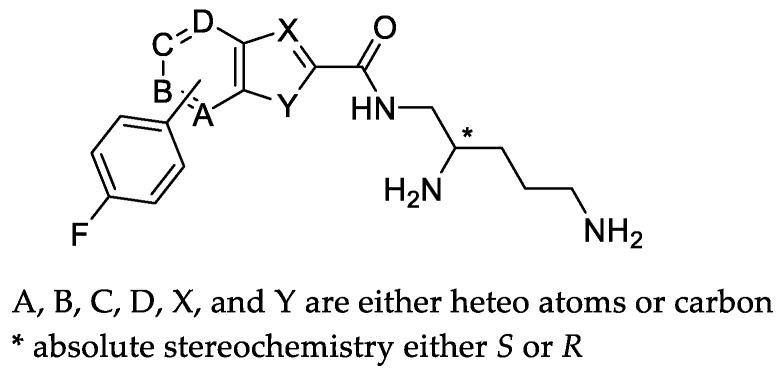
General structure of improved aryl hydrophobic head groups as potential EPIs.

**Figure 4 antibiotics-11-00030-f004:**
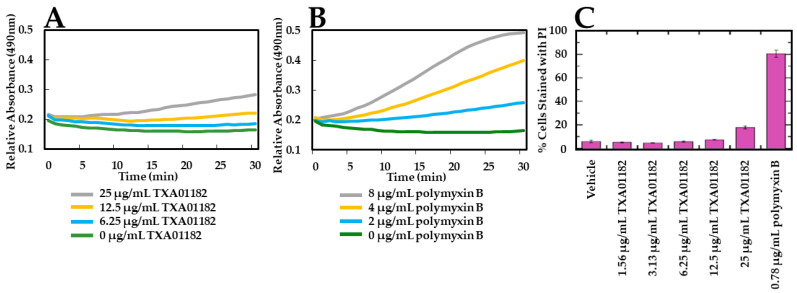
Outer- and inner-membrane permeabilization studies with TXA01182. Basal levels of NCF hydrolysis (**A**,**B**) or PI fluorescence (**C**) are observed upon addition of TXA01182 at concentrations below 12.5 and 25 μg/mL, respectively, indicating intact outer and inner membranes.

**Figure 5 antibiotics-11-00030-f005:**
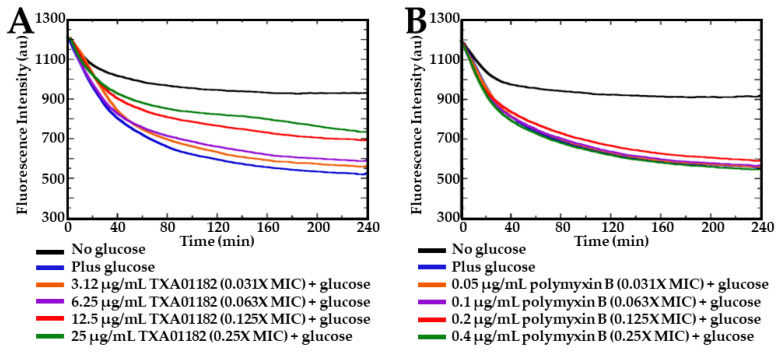
(**A**) TXA01182 concentration-dependent inhibition of EtBr efflux. (**B**) polymyxin B does not inhibit EtBr efflux under the same conditions.

**Figure 6 antibiotics-11-00030-f006:**
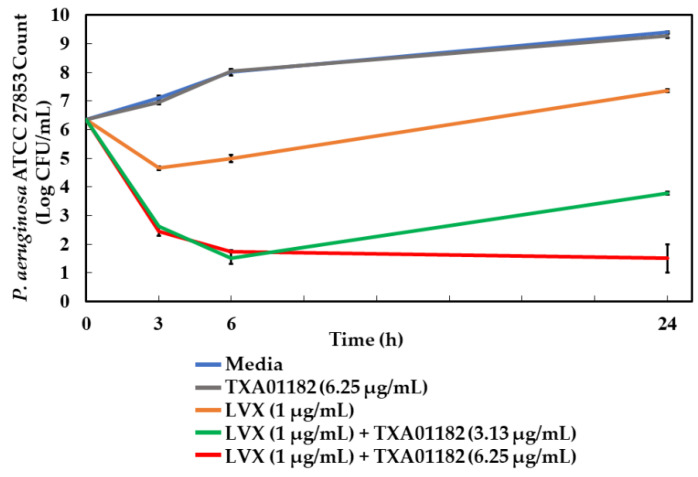
TXA01182 enhances the killing kinetics of levofloxacin (LVX). Time-kill kinetics of LVX alone and in combination with different concentrations of TXA01182 on *P. aeruginosa*. Values expressed as mean log_10_ of CFU/mL. Error bars represent standard deviation.

**Figure 7 antibiotics-11-00030-f007:**
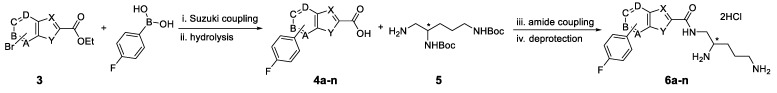
General synthetic scheme for compounds **6a**–**n**. Reagents and conditions: (i) Pd(dppf)Cl_2_, Na_2_CO_3_, toluene/EtOH/H_2_O, 100 °C; (ii) NaOH, THF/EtOH/H_2_O, r.t. to 60 °C; (iii) EDCI, HOBt, DIPEA, DMF, r.t.; (iv) 4 M HCl in dioxane/MeOH. * stereo centers.

**Figure 8 antibiotics-11-00030-f008:**
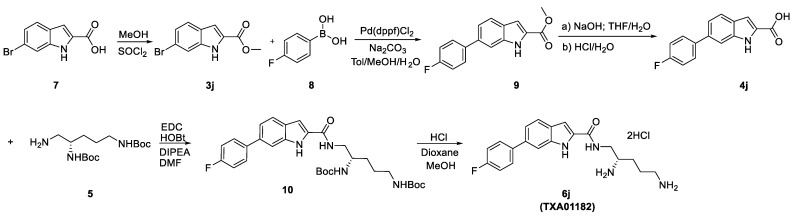
Synthetic scheme for compounds **6j** (TXA01182).

**Table 1 antibiotics-11-00030-t001:** Structures of screened compounds **6a**–**n**.

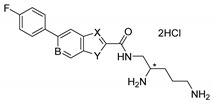						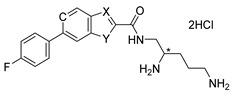				
Compound	X	Y	B	* Stereo		Compound	X	Y	C	* Stereo
**6a**	N	S	CH	*S*		**6b**	N	S	CH	*S*
**6c**	N	NH	CH	*S*		**6f**	CH	O	CH	*S*
**6d**	N	NH	CH	*R*		**6h**	CH	S	CH	*S*
**6e**	CH	O	CH	*S*		**6j (TXA01182)**	CH	NH	CH	*S*
**6g**	CH	S	CH	*S*		**6l**	CH	NH	CH	*R*
**6i**	CH	NH	CH	*S*		**6n**	CH	NH	N	*S*
**6k**	CH	NH	CH	*R*						
**6m**	CH	NH	N	*S*						

**Table 2 antibiotics-11-00030-t002:** Screening results of levofloxacin potentiation by fused heterocyclic carboxamides (6.25 µg/mL) in *P. aeruginosa* ATCC 27853 *.

Compound	6a	6b	6c	6d	6e	6f	6g	6h	6i	6j	6k	6l	6m	6n
MIC of EPI (µg/mL)	100	>100	100	>100	100	>100	25	25	100	100	>100	100	>100	>100
LVX MIC in the presence of EPI (µg/mL)	1	1	0.50	0.50	1	0.25	0.03	0.25	0.25	0.13	0.50	0.13	1	1
Fold Difference	1	1	2	2	1	4	32	4	4	8	2	8	1	1

* MIC of LVX against *P. aeruginosa* ATCC 27853 without EPI = 1 µg/mL.

**Table 3 antibiotics-11-00030-t003:** Potentiation of different classes of antimicrobials by TXA01182 against *P. aeruginosa* ATCC 27853.

Antimicrobial	MICs (µg/mL)		Fold Difference
	Alone	+ TXA01182 (6.25 µg/mL)	
Aztreonam	8	2	4
Cefepime	2	1	2
Ceftazidime	2	1	2
Azithromycin	64	32	2
Ciprofloxacin	0.25	0.125	2
Moxifloxacin	2	0.063	32
Levofloxacin	1	0.125	8
Cotrimoxazole	>256	16	>16
Doxycycline	32	2	16
Minocycline	32	1	32
Tigecycline	16	4	4
Chloramphenicol	>256	32	>8
Imipenem ^#^	4	4	1
Gentamicin ^#^	2	2	1

^#^ not the substrates for efflux pumps in *P. aeruginosa* [31].

**Table 4 antibiotics-11-00030-t004:** Levofloxacin potentiation comparison between TXA01182, MC-04,124, PAβN and CCCP on multidrug resistant clinical isolates of *P. aeruginosa*.

Strain	Levofloxacin MIC (μg/mL), (Fold Difference)					Resistance Mechanisms
	No EPI	+ TXA01182 (6.25 μg/mL)	+ MC-04,124 (6.25 μg/mL)	+ PAβN (50 μg/mL)	+ CCCP (12.5 μg/mL)	
AR-0229	64	4, (16)	64, (1)	64, (1)	64, (1)	*gyrA-T83I*, *nalC-G71E*, *mexR-V126Q*, *OXA-50*, *PAO*
AR-0239	64	8, (8)	64, (1)	8, (8)	64, (1)	*gyrA-T83I*, *nalC-G71E*, *mexR-V126Q*, *aac(6’)-IIa*, *aadB*, *aph(3’)-Ic*, *cmlA1*, *dfrB5*, *GES-1*, *OXA-10*, *OXA-50*, *strA*, *strB*, *tet(G)*, *VIM-11*
AR-0244	64	8, (8)	64, (1)	64, (1)	64, (1)	*gyrA-T133H*, *nalC-G71E*, *mexR-V126Q*, *OXA-50*
AR-0246	64	8, (8)	64, (1)	64, (1)	64, (1)	*gyrA-T83I*, *nalC-G71E*, *mexR-V126Q*, *aadB*, *NDM-1*, *OXA-10*, *OXA-50*, *PAO*, *rmtD2*, *tet(G)*, *VEB-1*
AR-0249	64	4, (16)	64, (1)	8, (8)	64, (1)	*gyrA-T83I*, *nalC-G71E*, *aac(3)-Id*, *aadA2*, *cmlA1*, *dfrB5*, *OXA-4*, *OXA-50*, *PAO*, *tet(G)*, *VIM-2*
AR-0264	64	4, (16)	64, (1)	64, (1)	64, (1)	*gyrA-D87Y*, *nalC-G71E*, *OXA-50*, *PAO*
AR-0232	8	0.5, (16)	ND	ND	8, (1)	*gyrA-T83I*, *nalC-G71E*, *mexR-V126Q*, *aadA6*, *OXA-50*, *PAO*, *strA*, *strB*, *sul1*, *tet(C)*
AR-0234	8	0.25, (32)	ND	ND	8, (1)	*gyrA-T83I*, *nalC-G71E*, *mexR-V126Q*, *aadA6*, *OXA-50*, *PAO*, *strA*, *strB*, *tet(C)*

The *nalC-G71E* mutation is associated with MexAB-OprM overexpression [42,43,44]. The *mexR-V126Q* mutation is associated with MexAB-OprM overexpression [45,46]. The *gyrA-T83I or gyrA-T133H* mutations lead to fluoroquinolone resistance [39,40,41]. ND: not determined.

**Table 5 antibiotics-11-00030-t005:** Frequency of resistance to TXA01182 and levofloxacin, alone and combination.

Strain	TXA01182 (6.25 μg/mL)	Levofloxacin (4 μg/mL)	Levofloxacin (4 μg/mL) + TXA01182 (6.25 μg/mL)
*P. aeruginosa* ATCC 27853	0.73	7.44 × 10^−8^	<1.30 × 10^−10^

## Data Availability

Data is contained within the article or Appendix A.

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
