# Peer review of "Evaluation of Heterocyclic Carboxamides as Potential Efflux Pump Inhibitors in Pseudomonas aeruginosa"

_antibiotics, 2021, doi:10.3390/antibiotics11010030_

Round 1
Reviewer 1 Report
Dear Authors,
The present study evaluates novel efflux pump inhibitors (EPIs) with significant potentiation in combination with levofloxacin against wild type Pseudomonas aeruginosa. The research subject is interesting and brings scientific important data in the field, as it deals with an important matter nowadays related to the fight against antimicrobial resistance. Some changes of the manuscript should nevertheless be performed in order to improve its quality. Following specific changes should thus be performed:
Minor changes
Please explain “ESKAPE pathogens” and “TAXISTANCETM commitment”.
Major changes
Introduction should contain data on similar studies existing in scientific literature and, in comparison, authors should emphasize the novelty and originality of their study. These informations need to be added in this part, but also in the Discussions section. Please clearly explain the aim of your study and give more details about previously performed studies, it is not at all clear what they consisted in.
References should be edited following the instructions for authors in the guide of the journal. Please check and edit.
Structure of the manuscript should be revised according to the Instructions for authors of the journal: Materials and Methods should be after Results and Discussion.
All these suggested changes should be performed in order to bring further improvements to the manuscript.
Author Response
We have revised the manuscript as per the reviewers’ comments. Please see below our point-by-point response to the first reviewer’s comments.
- Please explain “ESKAPE pathogens” and “TAXISTANCETMcommitment”
We have expanded the abbreviation for ESKAPE and removed the word “TAXISTANCE “from the manuscript.
- Introduction should contain data on similar studies existing in scientific literature and, in comparison, authors should emphasize the novelty and originality of their study. These informations need to be added in this part, but also in the Discussionssection. Please clearly explain the aim of your study and give more details about previously performed studies, it is not at all clear what they consisted in.
We have included new paragraphs in both introduction and in the discussion section to emphasize the aim of our study and details about previously performed studies.
- References should be edited following the instructions for authors in the guide of the journal. Please check and edit.
Done. The reference and the bibliography style are now as per MDPI requirement.
- Structure of the manuscript should be revised according to the Instructions for authors of the journal: Materials and Methodsshould be after Results and Discussion.
Materials and methods are now after Results and Discussion.
Reviewer 2 Report
In the article entitled "Evaluation of Heterocyclic Carboxamides as Potential Efflux Pump Inhibitors in Pseudomonas aeruginosa", Yuan and colleagues reported the design, synthesis and biological evaluation of aryl substituted heterocyclic carboxamides acting as efflux pump inhibitors (EPIs).
Eighteen newly designed compounds were synthesized and tested for their EP inhibition by several different biophysical and genetic assays, proving the MOA of the presented compounds.
I list below some annotations to the authors that should be addressed prior to publication.
- Row 84: "compounds" is written with a red s, it should be fixed;
- Figures 5-7: the chart legends should be reported outside the chart, in order to make it more clearly readable. Also, the quality of the same figures should be improved.
- In the Supplementary Materials (as declared also in the main text), the synthesis and chemical-physical characterization is reported for the sole TXA01182 derivative. If all the other compounds were already reported in the literature, the references with the synthetic pathway and full characterization should be reported. Differently, if the compounds were newly designed and synthesized, the detailed synthesis and full chemical-physical characterization of all the tested compounds should be listed in the Supplementary Materials file.
Author Response
We have revised the manuscript as per the reviewers’ comments. Please see below our point by point response to the Second reviewer’s comments.
- Row 84: "compounds" is written with a red s, it should be fixed;
- Figures 5-7: the chart legends should be reported outside the chart, in order to make it more clearly readable. Also, the quality of the same figures should be improved.
Done
- In the Supplementary Materials (as declared also in the main text), the synthesis and chemical-physical characterization is reported for the sole TXA01182 derivative. If all the other compounds were already reported in the literature, the references with the synthetic pathway and full characterization should be reported. Differently, if the compounds were newly designed and synthesized, the detailed synthesis and full chemical-physical characterization of all the tested compounds should be listed in the Supplementary Materials file.
The synthesis and the characterization of TXA01182 is now under materials and methods. The characterization of other newly designed and synthesized compounds is provided in the supporting information. The patent reference for some known compounds is provided.
Reviewer 3 Report
The manuscript described design, synthesis, and evaluation of novel efflux pump inhibitors. The authors synthesized many compounds. Among them, compound 6n, TXA01182, exhibited the most potent activity as an efflux pump inhibitor. Thus, these findings will be useful for anti-bacterial drug development. Therefore, the manuscript is not too excellent to be published. In other words, the manuscript is so excellent that it should be published.
Comments
(1) Which sites did synthesized efflux pump inhibitors bind to efflux pump proteins?
(2) Can synthesized efflux pump inhibitors be orally administered?
(3) Can compound 6n, TXA01182, show the inhibitory activity other than levofloxacin?
That is all.
Author Response
We have revised the manuscript as per the reviewers’ comments. Please see below our point-by-point response to the third reviewer’s comments.
- Which sites did synthesized efflux pump inhibitors bind to efflux pump proteins?
This piece of information is not available yet. Currently we are working on finding the binding sites of our EPIs.
- Can synthesized efflux pump inhibitors be orally administered?
Our Targe Product Profile (TPP) is IV-continuous infusion. Oral bioavailability of some selects compounds seem low.
- Can compound 6n, TXA01182, show the inhibitory activity other than levofloxacin?
Yes, it shows inhibitory activity with Aztreonam, all tetracyclines, all FQs, cotrimoxazole and Chloramphenicol.
Round 2
Reviewer 1 Report
Dear Authors,
The present study evaluates novel efflux pump inhibitors (EPIs) with significant potentiation in combination with levofloxacin against wild type Pseudomonas aeruginosa. The authors performed most of the suggested changes in the first round of review. Following specific changes should still be performed:
Minor changes:
Introduction still does not contain data on similar studies existing in scientific literature. Is this study singular or are there any similar studies in literature? Please clearly explain the aim of your study. I still don’t find it clear.
All these suggested changes should be performed in order to bring further improvements to the manuscript.
